# Exploring Fixed Point in Image Editing: Theoretical Support and Convergence Optimization

**Chen Hang[1]**[*], **Zhe Ma[2]**[*], **Haoming Chen[1], Xuwei Fang[3], Weisheng Xie[3],**
**Faming Fang[1]**[†] **Guixu Zhang[1]**, **Hongbin Wang[2]**[†]
[1]East China Normal University, Shanghai, 200062 China
[2]Ant Group, Hangzhou, 310063 China
[3]Bestpay AI Lab, Shanghai, 200080 China

## Abstract

In image editing, Denoising Diffusion Implicit Models (DDIM) inversion has become a widely adopted method and is extensively used in various image editing approaches. The core concept of DDIM inversion stems from the deterministic sampling technique of DDIM, which allows the DDIM process to be viewed as an Ordinary Differential Equation (ODE) process that is reversible. This enables the prediction of corresponding noise from a reference image, ensuring that the restored image from this noise remains consistent with the reference image. Image editing exploits this property by modifying the cross-attention between text and images to edit specific objects while preserving the remaining regions. However, in the DDIM inversion, using the $t-1$ time step to approximate the noise prediction at time step $t$ introduces errors between the restored image and the reference image. Recent approaches have modeled each step of the DDIM inversion process as finding a fixed-point problem of an implicit function. This approach significantly mitigates the error in the restored image but lacks theoretical support regarding the existence of such fixed points. Therefore, this paper focuses on the study of fixed points in DDIM inversion and provides theoretical support. Based on the obtained theoretical insights, we further optimize the loss function for the convergence of fixed points in the original DDIM inversion, improving the visual quality of the edited image. Finally, we extend the fixed-point based image editing to the application of unsupervised image dehazing, introducing a novel text-based approach for unsupervised dehazing.

## 1 Introduction

Diffusion models have gained significant attention in recent years. Early versions of Denoising Diffusion Probabilistic Models (DDPMs) [1] and score-based generative models [2] were capable of generating high-quality images and even surpassed the performance of Generative Adversarial Networks (GANs) [3] at that time. However, the initial diffusion models required a large number of sampling steps during image generation, which was a major bottleneck of diffusion models. Subsequent research focused on reducing the number of sampling steps without compromising the quality of generated images [4]. During this period, DDIM [5] was proposed and widely used for its speed and flexibility in both deterministic and stochastic generation. When DDIM is set to deterministic sampling, it can be viewed as an ODE process that is reversible. This property enables realistic image editing, leading to the introduction of Prompt-to-Prompt (P2P) [6] editing. The core idea of P2P is to modify the cross-attention between text and images. For local edits, such as replacing

---

[*]Equal contribution.
[†]Corresponding Author, fmfang@cs.ecnu.edu.cn, hongbin.whb@antgroup.com

38th Conference on Neural Information Processing Systems (NeurIPS 2024).

a cat with a dog while keeping the background unchanged, only the cross-attention corresponding to the cat needs to be modified. For enhancing certain effects, the cross-attention corresponding to the modifying word is reweighted. However, P2P still suffers from unreliable results in realistic image editing, as it exhibits inconsistencies even when directly reconstructing the reference image without any edits. The main cause of this issue lies in the errors introduced during DDIM inversion, which will be discussed later in this paper. To address this problem, some methods introduce auxiliary variables to mitigate the issue, such as null-text inversion (NTI) [7] for learning null-text embeddings. Exact diffusion inversion (EDICT) [8] uses two coupled vectors to invert each other in an alternating fashion, to achieve precise diffusion inversion. In recent methods, direct inversion (DI) [9] and inversion-free editing (InfEdit) [10] decouple the source and target branches to avoid optimizing DDIM inversion and achieve satisfactory results. However, these methods do not address the error issue in DDIM inversion theoretically. This issue can be formulated as solving fixed points of implicit functions, which has been mentioned in accelerated iterative diffusion inversion (AIDI) [11] and fixed-point inversion (FPI) [12]. However, AIDI and FPI only employ methods to solve fixed points of implicit functions without proving the existence of fixed points at each sampling step of DDIM inversion. Only FPI indirectly demonstrates the convergence of fixed-point loss through numerical experiments on the used dataset. Such numerical experiments cannot rigorously prove the existence of fixed points. Therefore, in this paper, we rigorously prove the existence and uniqueness of fixed points in DDIM inversion using the Banach fixed-point theorem [13]. Subsequently, based on this uniqueness, we identify flaws in the fixed-point loss used by AIDI and FPI and propose optimization. Finally, we explore the application of fixed-point based image editing in unsupervised image dehazing. In summary, our contributions in this paper are as follows:

- We theoretically prove that the Lipschitz constant in DDIM inversion is less than one. Based on the Banach fixed-point theorem, we establish the existence and uniqueness of fixed points, providing theoretical support for image editing methods that involve solving fixed points of implicit functions.

- Leveraging the uniqueness of fixed points, we demonstrate through theory and experimental cases that the existing methods suffer from flaws in the fixed-point loss and propose optimization.

- We extend the fixed-point based image editing approach to the task of unsupervised image dehazing and explore the feasibility of text-guided unsupervised dehazing through fixed-point based editing.

## 2 Preliminaries: DDPMs and DDIM Inversion

DDPMs model the probability distribution of a diffusion process by iteratively adding noise to a data sample until the data distribution becomes predominantly noise [1]. Subsequently, the data distribution is recovered from a random Gaussian noise through a reverse diffusion process. The diffusion process is typically regarded as a Markov chain starting from $x_0$ and obtaining $x_1, x_2, ..., x_T$ by adding noise. The specific process is outlined as follows:

$$q\left(x_{1:T} \mid x_0\right) := \prod_{t=1}^{T} q\left(x_t \mid x_{t-1}\right), \quad q\left(x_t \mid x_{t-1}\right) := \mathcal{N}\left(x_t; \sqrt{1 - \beta_t} x_{t-1}, \beta_t I\right), \quad (1)$$

where schedule $\beta_0, \beta_1, ..., \beta_T \in (0,1)$. $x_t$ can be expressed as a linear combination of $x_0$ and Gaussian noise $\epsilon \sim \mathcal{N}(0, I)$:

$$x_t = \sqrt{\bar{\alpha}_t} x_0 + \sqrt{1 - \bar{\alpha}_t} \epsilon, \quad \alpha_t := 1 - \beta_t, \quad \bar{\alpha}_t := \prod_{s=1}^{t} \alpha_s, \quad (2)$$

In DDPMs, both the diffusion process and the reverse diffusion process are Markov chains, which require a large number of sampling steps during image generation. However, DDIM introduces a method to accelerate sampling by decoupling the reverse diffusion process from the Markov chain, making it feasible to perform skip-step sampling and reducing the number of sampling steps [5]. Simultaneously, the introduction of Stable Diffusion (SD) [14] alleviates the resource requirements for training diffusion models and improves inference speed. SD involves compressing the image

into a latent space for diffusion instead of operating in the original pixel-level space. As a result, the reverse diffusion process can be expressed as follows:

$$z_{t-1} = \sqrt{\frac{\bar{\alpha}_{t-1}}{\bar{\alpha}_t}} z_t + \left( \sqrt{1 - \bar{\alpha}_{t-1}} - \sqrt{\frac{(1 - \bar{\alpha}_t)\bar{\alpha}_{t-1}}{\bar{\alpha}_t}} \right) \epsilon_\theta \left( z_t, t, p \right), \tag{3}$$

where $\epsilon_\theta$ represents a network that predicts added noise, $z_t$ denotes the compressed representation of $x_t$ in the latent space, $t$ represents the time step, and $p$ represents the encoding of a condition text.

Now let's focus on DDIM inversion. DDIM inversion seeks to find a noise $z_T$ given the latent space representation $z_0$ corresponding to a given $x_0$ and the associated textual prompt $p$. This allows for the autoregressive reconstruction of $z_0$. The process of DDIM inversion can be rewritten by Equation 3 as follows:

$$z_t = \sqrt{\frac{\bar{\alpha}_t}{\bar{\alpha}_{t-1}}} z_{t-1} + \left( \sqrt{1 - \bar{\alpha}_t} - \sqrt{\frac{(1 - \bar{\alpha}_{t-1})\bar{\alpha}_t}{\bar{\alpha}_{t-1}}} \right) \epsilon_\theta \left( z_t, t, p \right) \tag{4}$$

$$\approx \sqrt{\frac{\bar{\alpha}_t}{\bar{\alpha}_{t-1}}} z_{t-1} + \left( \sqrt{1 - \bar{\alpha}_t} - \sqrt{\frac{(1 - \bar{\alpha}_{t-1})\bar{\alpha}_t}{\bar{\alpha}_{t-1}}} \right) \epsilon_\theta \left( z_{t-1}, t, p \right)$$

Therefore, we can invert $z_0$ back to $z_T$ using Equation 4, allowing us to reconstruct $z_0$ using the reverse diffusion process. However, it can be observed that in Equation 4, $z_t$ is directly approximated by $z_{t-1}$, which inevitably introduces errors that accumulate over time. This inconsistency between the forward and reverse processes of the diffusion leads to poor quality in image reconstruction and editing [7, 8].

## 3   Theoretical Support for Fixed Point in DDIM Inversion

From the first equation of Equation 4, it is evident that finding $z_t$ can be regarded as solving a fixed-point problem for the implicit function with respect to $\epsilon_\theta \left( z_t, t, p \right)$. Therefore, we can consider the right-hand side of the equation as a function with $z_t$ as a variable:

$$f(z_t) = \sqrt{\frac{\bar{\alpha}_t}{\bar{\alpha}_{t-1}}} z_{t-1} + \left( \sqrt{1 - \bar{\alpha}_t} - \sqrt{\frac{(1 - \bar{\alpha}_{t-1})\bar{\alpha}_t}{\bar{\alpha}_{t-1}}} \right) \epsilon_\theta \left( z_t, t, p \right), \tag{5}$$

Then, to find $f(z_t) = z_t$, we can use $z_{t-1}$ as the starting point for iteration. The iteration process can be expressed as follows:

$$z_t^0 = z_{t-1}, \quad z_t^N = z_t, \quad z_t^{i+1} = f\left( z_t^i \right), \quad (i = 0, 1, ..., N) \tag{6}$$

After multiple iterations, the fixed point of Equation 5 can be found. Now, we focus on the theoretical analysis of whether a fixed point exists in Equation 5. Let $z_t^i$ and $z_t^j$ represent any two points in the function $f(z_t)$ at time step $t$, where $i$ and $j$, with $j$ greater than $i$, are within the range of 0 to $N$. We can obtain the following:

$$||f(z_t^i) - f(z_t^j)|| = \left( \sqrt{1 - \bar{\alpha}_t} - \sqrt{\frac{(1 - \bar{\alpha}_{t-1})\bar{\alpha}_t}{\bar{\alpha}_{t-1}}} \right) ||\epsilon_\theta(z_t^i, t, p) - \epsilon_\theta(z_t^j, t, p)|| \tag{7}$$

$$\leq \sqrt{1 - \bar{\alpha}_t} ||\epsilon_\theta(z_t^i, t, p) - \epsilon_\theta(z_t^j, t, p)||$$

where $|| \cdot ||$ is $L_2$ norm. We know that in the reverse diffusion process, when predicting the $z$ value at time step $t - 1$ given the $z$ value at time step $t$, an initial rough estimation of $z_0$ is made. This rough estimation of $z_0$ introduces some errors compared to the final obtained value of $z_0$. However, the overall shape of the image is already quite similar, and as time $t$ decreases, this rough estimation of $z_0$ gradually approaches the final obtained $z_0$. This rough estimation of $z_0$ can be represented as follows:

$$z_0^i = \left( z_t^i - \sqrt{1 - \bar{\alpha}_t}\epsilon_\theta(z_t^i, t, p) \right) / \sqrt{\bar{\alpha}_t}, \quad z_0^j = \left( z_t^j - \sqrt{1 - \bar{\alpha}_t}\epsilon_\theta(z_t^j, t, p) \right) / \sqrt{\bar{\alpha}_t}, \tag{8}$$

where $z_0^i$ and $z_0^j$ represent the predictions of time step zero. Although $z_0^i$ and $z_0^j$ obtained from $z_t^i$ and $z_t^j$ are very close, there might be some errors when $t$ is not close to zero. Let's denote this error as $\varepsilon_0$. Additionally, let's define the difference between $z_t^i$ and $z_t^j$ as $\varepsilon_t$. With Equation 8, we can derive the following relationship:

$$z_t^i - z_t^j = \sqrt{\bar{\alpha}_t}\varepsilon_0 + \sqrt{1 - \bar{\alpha}_t}\left(\epsilon_\theta(z_t^i, t, p) - \epsilon_\theta(z_t^j, t, p)\right), \tag{9}$$

Due to the fact that different initial values at time step $t$ can lead to similar $z_0$, i.e., any position at time step $t$ can converge to a small range defined by the rough estimation of $z_0$, we can obtain $|\varepsilon_0| = k|\varepsilon_t|$, where $0 < k < 1$. Moreover, from the perspective of ODE, the paths from $z_t^i$ to $z_0^i$ and from $z_t^j$ to $z_0^j$ can be viewed as two ODE trajectories with initial values of $z_t^i$ and $z_t^j$, respectively, or as two integral curves of ODE given the initial values $z_t^i$ and $z_t^j$. These integral curves non-tangent and non-intersecting [15]. Therefore, we can conclude that $\varepsilon_0$ and $\varepsilon_t$ have the same sign. We can also prove this by contradiction: if $\varepsilon_0$ and $\varepsilon_t$ have opposite signs, then the two ODE trajectories would have an intersection point. From that intersection point, there would exist two different paths in both forward and backward directions, which contradicts the deterministic sampling of DDIM. Consequently, we can deduce $\varepsilon_0 = k\varepsilon_t$, where $0 < k < 1$. By substituting this into Equation 9, we obtain:

$$z_t^i - z_t^j = k\sqrt{\bar{\alpha}_t}\varepsilon_t + \sqrt{1 - \bar{\alpha}_t}\left(\epsilon_\theta(z_t^i, t, p) - \epsilon_\theta(z_t^j, t, p)\right) \tag{10}$$
$$= k\sqrt{\bar{\alpha}_t}\left(z_t^i - z_t^j\right) + \sqrt{1 - \bar{\alpha}_t}\left(\epsilon_\theta(z_t^i, t, p) - \epsilon_\theta(z_t^j, t, p)\right)$$

By rearranging the Equation 10 and substituting it into Equation 7, we obtain:

$$||f(z_t^i) - f(z_t^j)|| \le \left(1 - k\sqrt{\bar{\alpha}_t}\right)||z_t^i - z_t^j||, \tag{11}$$

From Equation 2, it is known that $\bar{\alpha}_t \in (0, 1)$. Therefore, it can be inferred that $(1 - k\sqrt{\bar{\alpha}_t}) \in (0, 1)$. This proves that the implicit function $f(z_t)$ satisfies the contraction mapping condition. By further applying the Banach fixed-point theorem [13], we can conclude that the fixed point of $f(z_t)$ exists and is unique.

## 4 Convergence Optimization for Fixed Point in DDIM Inversion

In AIDI [11] and FPI [12], to determine the convergence of the fixed point obtained through iterative processes in the implicit function $f(z_t)$, the criterion used is $f(z_t) - z_t$. In FPI, numerical experiments were conducted on the given dataset, and it was concluded that convergence is achieved within $2 \sim 3$ iterations. Furthermore, in Section 3, we have proved the uniqueness of the fixed point. Therefore, at any position with a time step $t$, convergence can be reached towards the same point. We express this using Equation 11 as follows:

$$||f(z_t^i) - f(z_t^j)|| \le \left(1 - k\sqrt{\bar{\alpha}_t}\right)||z_t^i - z_t^j||,$$
$$||f(\hat{z}_t^i) - f(\hat{z}_t^j)|| \le \left(1 - k\sqrt{\bar{\alpha}_t}\right)||\hat{z}_t^i - \hat{z}_t^j||, \tag{12}$$
$$||f(z_t^j) - f(\hat{z}_t^j)|| \le \left(1 - k\sqrt{\bar{\alpha}_t}\right)||z_t^j - \hat{z}_t^j||,$$

where $\hat{z}_t^i = z_t^i + \epsilon, \epsilon \sim \mathcal{N}(0, I)$, and $\hat{z}_t^j$ is the result obtained by iterating $\hat{z}_t^i$ in the implicit function $f(z_t)$. It can also be considered that $\hat{z}_t^i$ is a new path distinct from $z_t^i$. Theoretically, the three cases of Equation 12 have the same convergence rate, which means they have the same Lipschitz constant. However, in practical computations, it is challenging for two different paths to converge exactly to the same fixed point, resulting in numerical errors. Consequently, this leads to the persistence of numerical discrepancies in the third inequality of Equation 12 even at the end of the iteration.

$$||f(z_t^N) - f(\hat{z}_t^N)|| > \varepsilon, \quad \varepsilon > 0, \tag{13}$$

Incorporating the error into the third inequality of Equation 12 for correction, we have:

$$||f(z_t^j) - f(\hat{z}_t^j)|| \le \left(1 - k\sqrt{\bar{\alpha}_t}\right)||z_t^j - \hat{z}_t^j|| + \varepsilon \tag{14}$$
$$\le \left(1 - k\sqrt{\bar{\alpha}_t} + \delta\right)||z_t^j - \hat{z}_t^j||, \quad \delta > 0$$

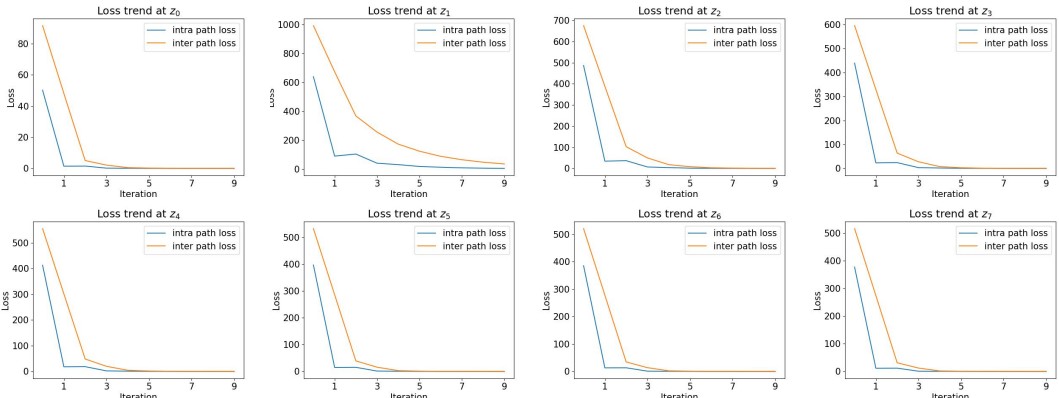

Figure 1: In the diffusion model with a step size of $50$ steps, we selected the trend of fixed-point loss for the first $8$ steps, while the trends for the remaining steps were consistent and are provided in the appendix A.1. In each step, we performed $10$ iterations. The blue intra path loss represents $f(z_t) - z_t$ used in AIDI and FPI, while the orange inter path loss represents our optimized $f(z_t) - f(\hat{z}_t)$. The trend of the intra path loss aligns with the findings reported in FPI, converging within $2 \sim 3$ iterations. However, the inter path loss exhibits noticeable lag. Based on the proven uniqueness of fixed points in this paper, the convergence of the intra path loss does not indicate the convergence of fixed points. Instead, attention should be paid to the inter path loss to assess the convergence of fixed points.

It is not difficult to observe that if there are multiple paths and each path only focuses on the variation of the loss within the path, they will have the same convergence rate. However, when considering the loss between paths, there will be a noticeable lag in the convergence rate, as shown in Figure 1. The loss function $f(z_t) - z_t$ used in AIDI and FPI precisely corresponds to the case of solely focusing on the variation within paths. As a result, the fixed points obtained when the $f(z_t) - z_t$ loss converges may not be the optimal points. To optimize the original criterion for judging the convergence of fixed points, we modify it to $f(z_t) - f(\hat{z}_t)$ and define the convergence of the $f(z_t) - f(\hat{z}_t)$ loss as the convergence of fixed points. Through experimentation, we have also confirmed that the convergence of fixed points actually lags behind what is mentioned in FPI, as illustrated in Figure 1. In order to visually demonstrate the process of fixed point convergence, we randomly select two points from $z_t$ as the axes. Subsequently, multiple paths are sampled by applying $\hat{z}_t^i = z_t^i + \epsilon, \epsilon \sim \mathcal{N}(0, I)$. The results obtained from multiple iterations are plotted as trajectory lines, as shown in Figure 2.

By optimizing the fixed point convergence criterion and the fixed-point loss, the computed fixed points will be more accurate. To further improve the precision of fixed points, it is natural, as suggested by Figure 2, to consider selecting the final points from multiple paths, finding the cluster centers of these points, and using them as the results of the fixed points. However, this approach would increase the computational burden. Therefore, in subsequent experiments, we only consider the results obtained when the inter path loss converges.

## 5   Generalization of Fixed Point in DDIM Inversion to Image Dehazing

Image dehazing, as a subtask of low-level vision, primarily aims to remove the haze degradation from hazy images to recover clean images, thereby assisting high-level visual tasks [16, 17] for improved performance. Hazy images are typically modeled as follows [18]:

$$I(x) = J(x)t(x) + A(1 - t(x)), \tag{15}$$

where $I(x)$ is hazy image, $J(x)$ is the real scene to be recovered. $t(x) = e^{-kd(x)}$ is the medium transmission, where $d(x)$ is depth and $k$ is scattering coefficient of the atmosphere. $A$ is the global atmospheric light.

Existing unsupervised image dehazing methods can be broadly categorized into learning-based and non-learning-based approaches. Learning-based methods primarily utilize unpaired images [19, 20, 21] to train networks for unsupervised dehazing, while non-learning-based methods rely on image priors such as dark channel prior [22] and rank-one prior [23]. With the development of visual

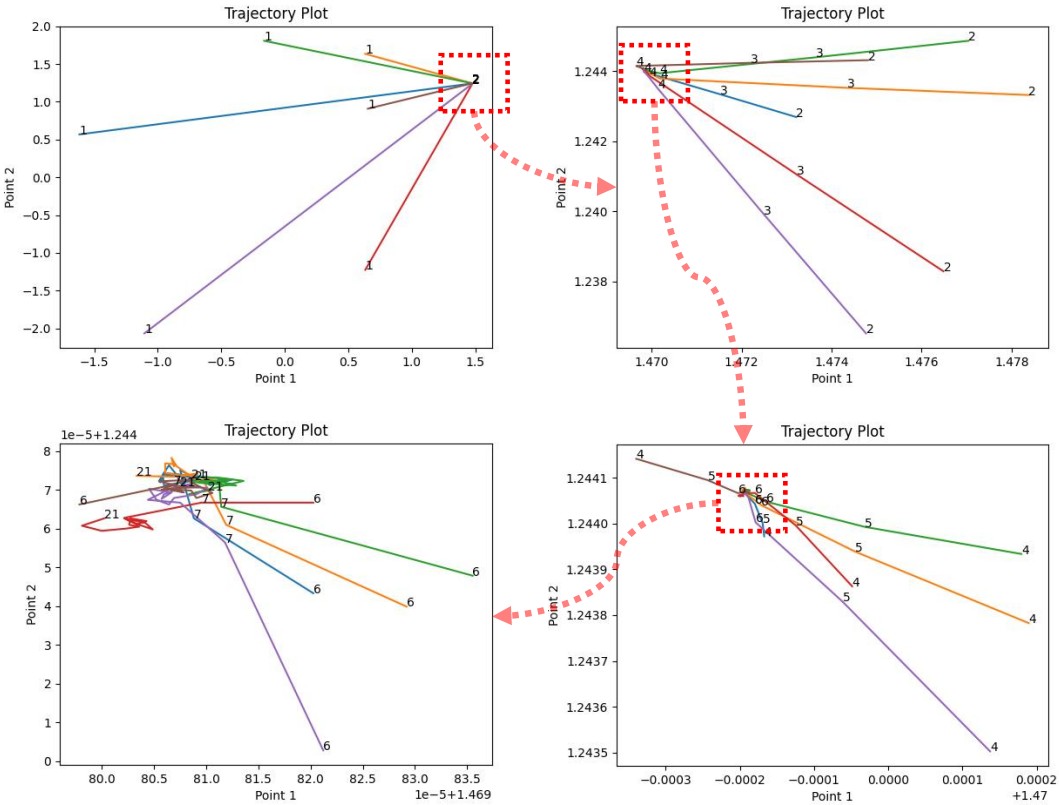

Figure 2: The axes were randomly chosen from two points in $z_t$. The total number of iterations to compute the fixed points was 21, and the corresponding step numbers were labeled in each path in each subgraph. It can be observed that the inter path loss starts to converge around the 7th step and beyond, indicating that the convergence of fixed points requires approximately 7 iterations.

language models, some supervised multimodal dehazing approaches have emerged [24]. However, there are few reports on unsupervised multimodal dehazing. This paper mainly explores image-editing-based unsupervised dehazing, where the main principle is to achieve dehazing effects through cross-attention replacement of corresponding to haze. Since the haze is replaced with null text, the null text optimization from NTI is necessary [7]. Otherwise, as the class-free guidance coefficient increases, the recovered image may collapse. The DDIM Inversion used in NTI does not involve fixed point correction. Therefore, in subsequent experiments, we will show that the use of fixed-point optimized NTI can effectively alleviate the image collapse issue in dehazed images.

# 6 Experiments

This section will be divided into two parts. The first part focuses on image editing experiments, aiming to highlight that modifying the convergence criterion of fixed points leads to better visual quality of the edited image compared to previous approaches. Additional techniques have also been introduced in AIDI and FPI to further enhance the visual effects of image editing. However, to avoid interference, we only reproduced their fixed point code. For the dataset, we selected PIE-Bench [9] in the context of image editing, which consists of 700 images featuring 10 distinct editing types.

The second part comprises experiments on unsupervised image dehazing. Our experiments are based on the NTI approach, incorporating fixed points to alleviate the issue of image collapse in NTI based dehazed images. We directly utilized the real-world scenes provided in the RESIDE [25] dataset for our experiments. Finally, both parts of the experiment employed identical configurations, comprising a sampling step of 50 and a class-free guidance coefficient of 7.5. In the context of image dehazing, to preclude any potential disruption of haze-free regions, no additional text descriptions were introduced. The original prompt was solely 'haze', while the target prompt was designated as '$\emptyset$'.

| Input | P2P | $f(z_t) - z_t$ | $f(z_t) - f(\hat{z}_t)$ |

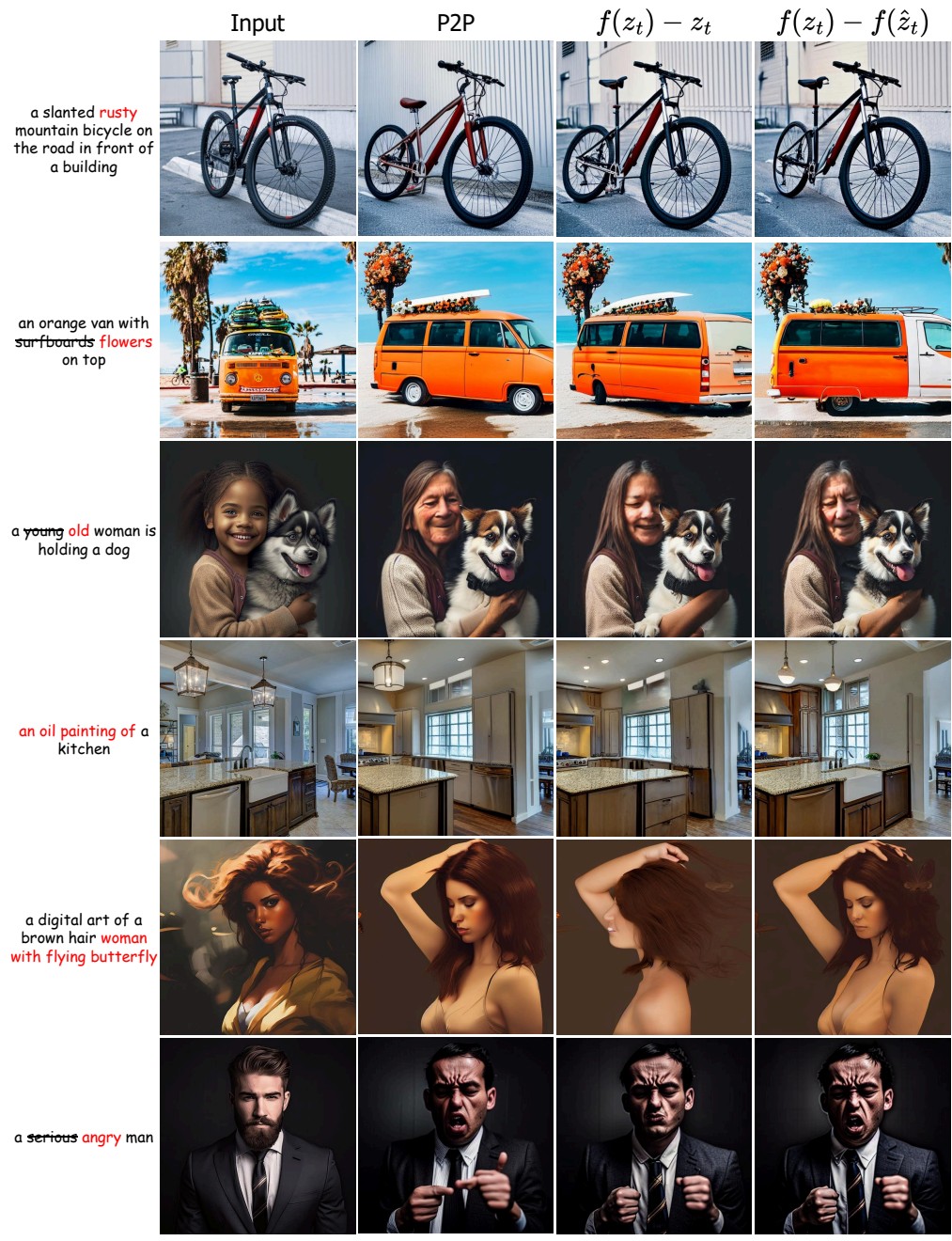

Figure 3: $f(z_t) - z_t$ denotes the intra path loss, while $f(z_t) - f(\hat{z}_t)$ signifies the optimized inter path loss. In the prompt, red signifies additions, whereas strikethrough indicates deletions. More results can be found in appendix A.2.

## 6.1 Image Editing

In the experiments on image editing, we primarily focused on the P2P and its variations: P2P with fixed points and P2P with improved convergence of fixed points. P2P serves as a baseline to demonstrate the effectiveness of fixed points. The proposed optimization for fixed point convergence is compared to the original fixed-point method to showcase the improved visual quality of the edited images. We evaluate the effectiveness of our proposed optimization from two perspectives. Firstly, we directly compare the imaging results of image editing, as shown in Figure 3 and Table 1. In

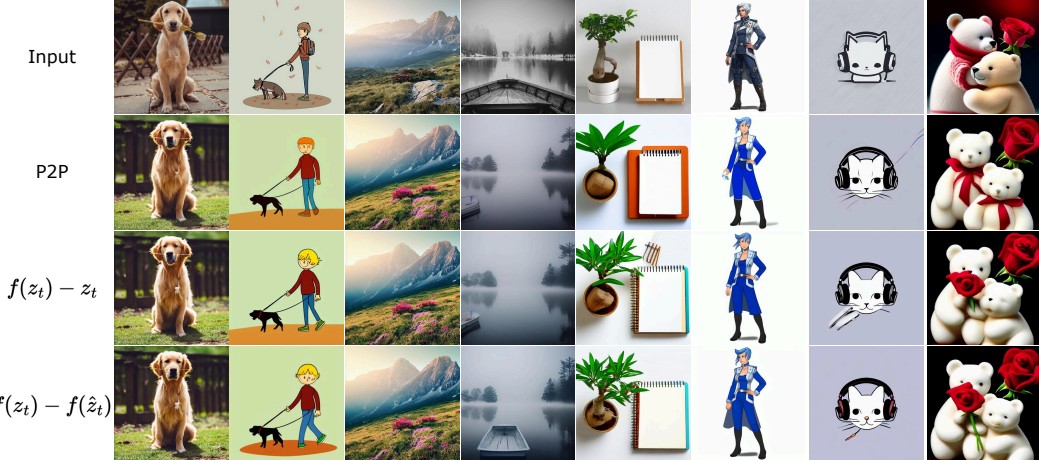

Figure 4: The autoregressive reconstruction result derived from noise subjected to DDIM inversion.

Table 1: Performance results for image editing and reconstruction, with evaluation metrics drawn from PIE-Bench [9], are presented across seven indicators spanning three dimensions. **Bold** indicate the best results.

| Method | Structure | Background Preservation | | | | CLIP Similariy | |
|---|---|---|---|---|---|---|---|
| | Distance$_{\times}10^3 \downarrow$ | PSNR $\uparrow$ | LPIPS$_{\times}10^3 \downarrow$ | MSE$_{\times}10^4 \downarrow$ | SSIM$_{\times}10^2 \uparrow$ | Whole $\uparrow$ | Edited $\uparrow$ |
| | Image Editing / Image Reconstruction | | | | | | |
| P2P | 69.953 / 70.753 | 17.87 / 17.73 | 208.89 / 210.78 | 219.88 / 225.78 | 71.14 / 70.91 | 25.01 / 23.69 | 22.44 / 21.29 |
| $f(z_t) - z_t$ | 69.521 / 70.495 | 17.83 / 17.73 | 208.98 / 210.76 | 221.48 / 225.60 | 71.14 / 70.91 | 25.19 / 23.66 | 22.53 / 21.33 |
| $f(z_t) - f(\hat{z}_t)$ | **69.190 / 70.046** | **17.87 / 17.74** | **208.88 / 210.71** | **219.85 / 225.50** | **71.15 / 70.91** | **25.31 / 23.80** | **22.58 / 21.37** |

Figure 3, we selected some results that exhibit noticeable differences before and after optimization. It can be observed that optimizing the convergence of fixed points leads to improved performance and robustness. Secondly, we also evaluate the three methods in terms of autoregressive reconstruction of the images, as shown in Figure 4 and Table 1. It can be observed that using the optimized fixed points yields reconstructed images that have higher similarity to the input images.

## 6.2 Image Dehazing

In the unsupervised image dehazing experiments, we primarily compared the NTI approach with NTI incorporating fixed points. From the experimental results depicted in Figure 5 and Table 2, it can be observed that the inclusion of fixed points in NTI effectively mitigates the image collapse issue in certain dehazed images. As mentioned earlier, in unsupervised dehazing, where the attention for haze is replaced with null text attention, not using NTI would result in image collapse when the class-free guidance coefficient increases. Therefore, we also included examples of image collapse in Figure 5.

Table 2: Results of dehazing in real-world scenes, evaluation was conducted utilizing Non-Reference Image Quality Assessment (NR-IQA). **Bold** indicate the best results.

| Method | NR-IQA | |
|---|---|---|
| | BRISQUE $\downarrow$ | NIQE $\downarrow$ |
| NTI | 27.5879 | 4.4499 |
| NTI w fixed point | **24.1297** | **3.4777** |
| fixed point w/o NTI | 37.1827 | 7.7027 |

## 7 Limitations, Prospects and Impact

Although the fixed-point loss achieves better visual quality after optimization, there is still room for improvement. Strategies proposed in AIDI and FPI can be employed in image editing to further enhance the consistency between edited images and reference images. However, in this study, we did not reproduce the mentioned strategies in order to avoid interference with the optimization of the fixed-point. We now primarily discuss the limitations of unsupervised image dehazing and other image restoration tasks. In unsupervised dehazing, although some haze can be effectively removed

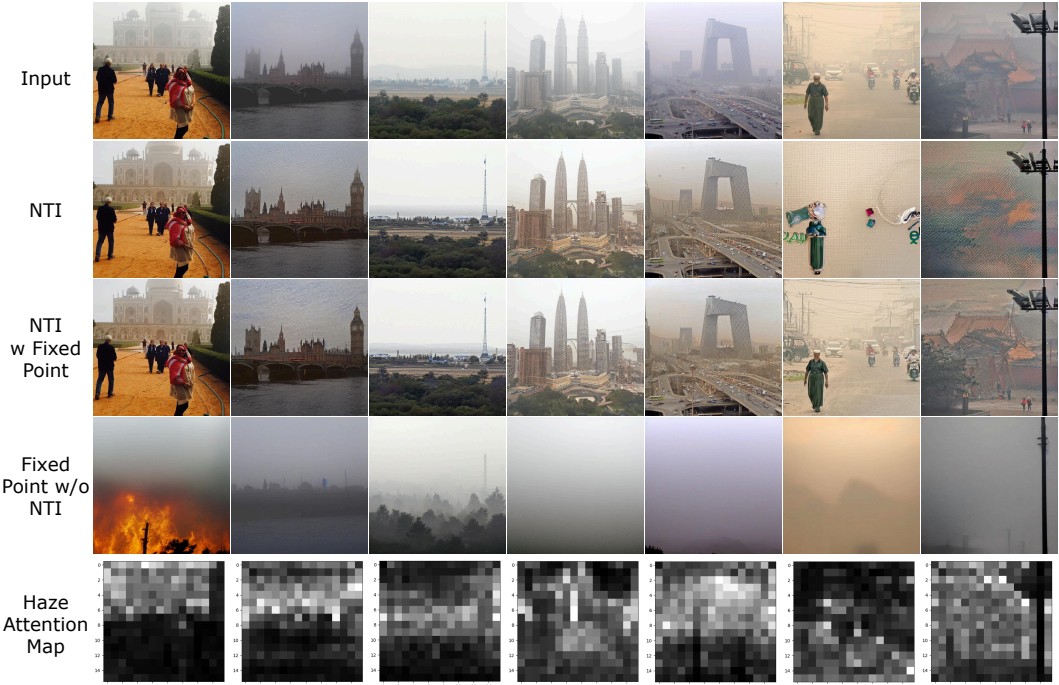

Figure 5: Imaging results of image dehazing under real-world scenes in RESIDE dataset.

from images, there are still issues with artifacts. For example, in the middle column of Figure 5, some artifacts can be observed. We believe that one reason is the imprecise attention map corresponding to the haze. We present the matched attention map in last line from Figure 5. It can be seen that the attention map corresponding to the haze semantics only provides a rough match, far from pixel-level alignment. After analyzing the process of attention map generation, we found that the attention map mainly resides in the deeper layers of the Unet. This implies that pixel-level fine-grained alignment cannot be achieved, which also hinders the direct transfer of image editing to some fine-grained image restoration tasks such as image deraining.

We have summarized some experiences in attempting to improve the performance of unsupervised image dehazing. A more straightforward approach is to fine-tune the dehazing task using existing mature techniques such as Lora [26] and Adapter [27], enabling the model to match more accurate haze attention maps. Another strategy that does not require training is to introduce image priors to weight the haze attention map. Since the attention map is primarily obtained by summing and averaging multiple layers of cross-attention, the priors need to be weighted into each layer of cross-attention. However, the cross-attention at different layers may have little correlation with the final haze attention map in terms of morphology. We attempted to directly weight the dark channel prior [22] and depth estimation prior [28] into each layer of cross-attention and found that this direct weighting method can improve the restoration results of some images but may also cause some images to collapse (see appendix A.3). Therefore, designing a weighted algorithm specifically for priors is necessary. In addition, similar to other generative models [29, 30, 31], the ability of image editing can be exploited to generate deceptive and harmful content, and the fixed-point based image editing may potentially exacerbate the negative impact of deep generative models for malicious purposes.

## 8 Conclusion and Future Work

This paper fills the theoretical gap in the fixed-point theory of DDIM inversion for image editing, providing theoretical support for image editing methods that employ fixed points. Additionally, based on the unique conclusion regarding the existence of fixed points obtained in this paper, we optimize the convergence criterion for fixed points and propose the inter path loss as a measure for assessing fixed-point convergence, supported by both theoretical analysis and experimental results. Through

experiments conducted on PIE-Bench, we demonstrate improved visual quality after optimization. Finally, we extend the application of fixed-point based image editing to unsupervised image dehazing and analyze the limitations and prospects of image restoration tasks. In our future works, we will delve further into fixed-point computations and unsupervised semantic based image restoration. Regarding fixed points, our exploration will focus on reducing computational overhead associated with their calculation and enhancing the precision. Concerning unsupervised image restoration, we will primarily examine ways to improve the corrective impact of priors on imprecise semantic attention maps, thereby achieving superior image restoration outcomes.

## 9 Acknowledgements

This work was supported by the National Key R&D Program of China (2022ZD0161800), and the National Natural Science Foundation of China under Grant 62271203. Besides, this work were supported by Ant Group and Bestpay AI Lab.

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

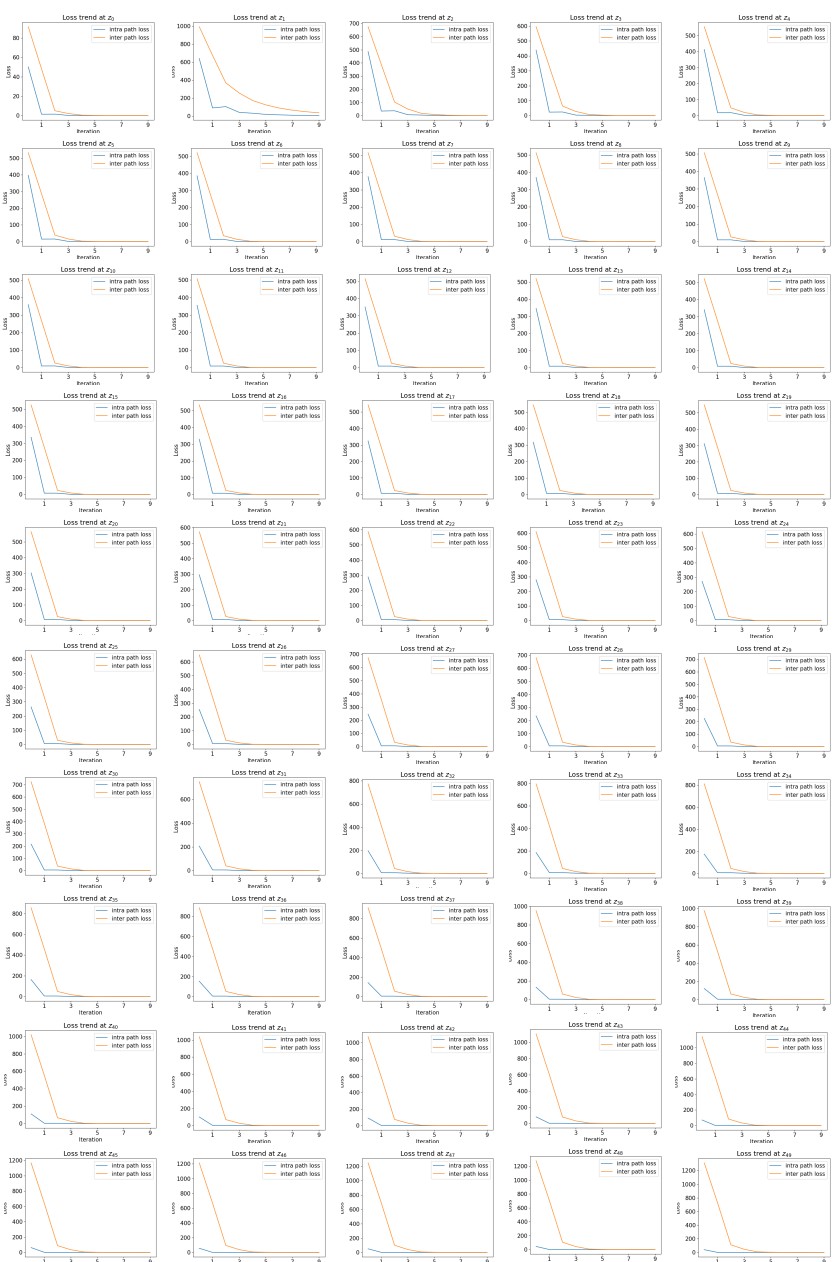

Figure 6: The trend of fixed-point loss for 50 steps.

# A  Appendix

## A.1  Loss Trend for 50 steps

We provide the complete loss trend in Figure 6. It can be observed that the intra path loss exhibits noticeable hysteresis. Since the primary purpose is to demonstrate the hysteresis rather than illustrate the final convergence position, we did not use logarithmic scaling on the coordinate axes.

## A.2  More Experiments

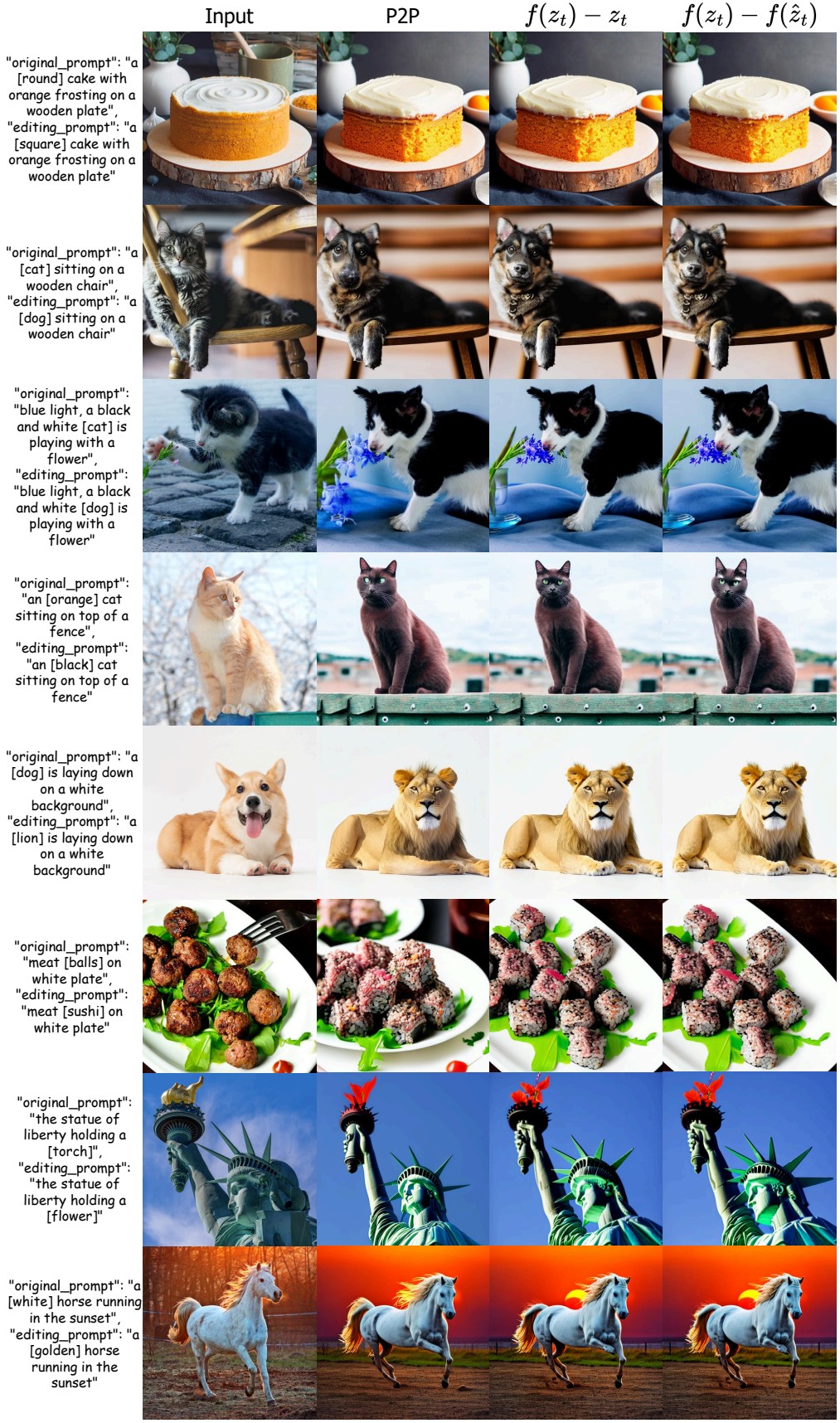

"original_prompt": "
[black] chair in a
conference room",
"editing_prompt": "
[blue] chair in a
conference room"

"original_prompt": "a
fluffy dog with a
blue leash sitting in
the [grass]",
"editing_prompt": "a
fluffy dog with a
blue leash sitting in
the [ground]"

"original_prompt": "a
[meerkat] puppy
wrapped in a blue
towel",
"editing_prompt": "a
[lion] puppy wrapped
in a blue towel"

"original_prompt": "
[purple] tulips in
vase",
"editing_prompt": "
[yellow] tulips in
vase"

"original_prompt": "a
woman in a [jacket]
standing in the rain",
"editing_prompt": "a
woman in a [blouse]
standing in the rain"

"original_prompt":
"the city of dresden,
germany, europe",
"editing_prompt": "[a
sunny day of] the
city of dresden,
germany, europe"

"original_prompt": "a
cute dog holding a
[red] heart",
"editing_prompt": "a
cute dog holding a
[pink] heart"

"original_prompt": "a
water droplet hangs
from a string of
lights",
"editing_prompt": "a
water droplet hangs
from a string of
[red] lights"

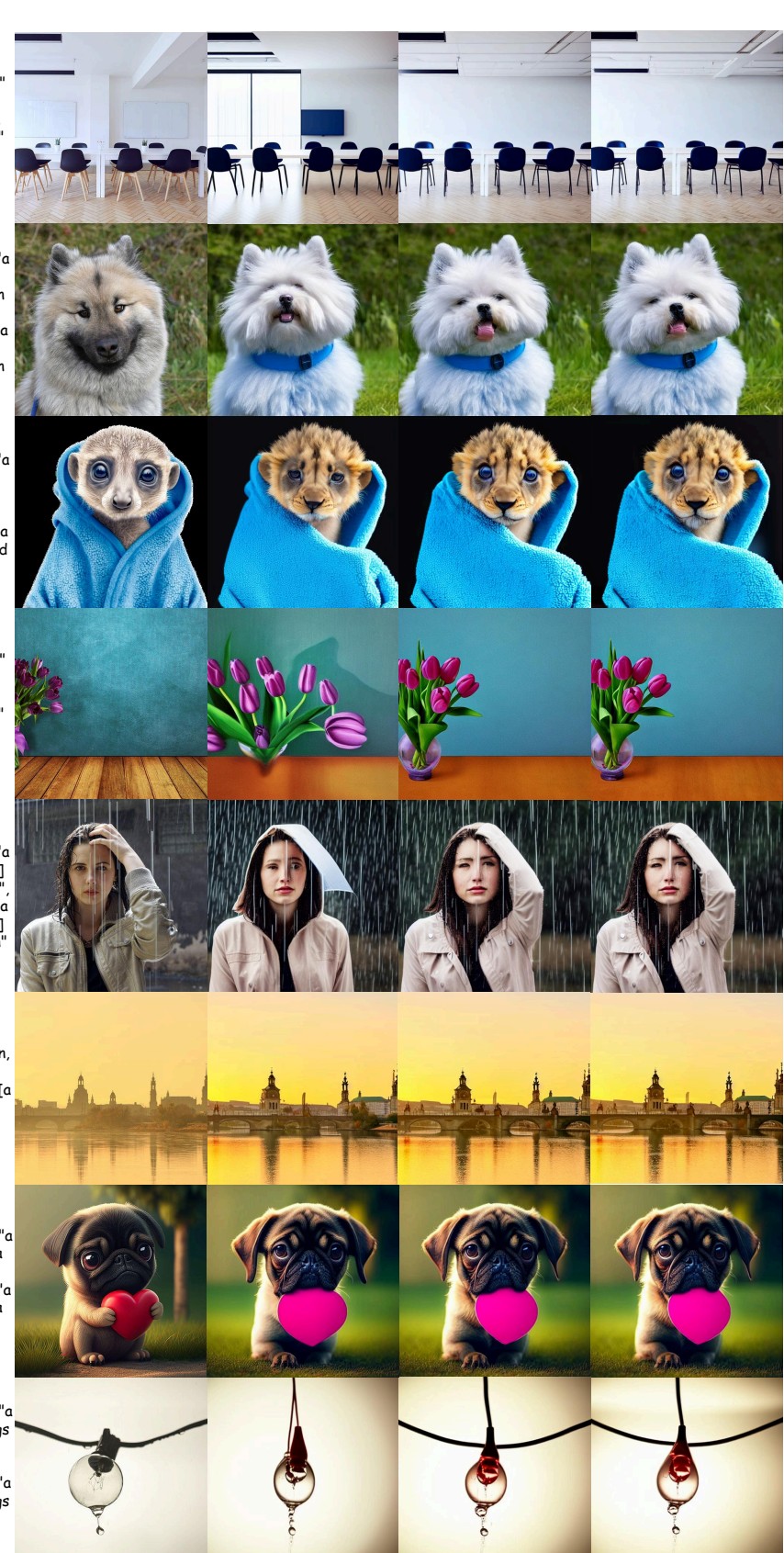

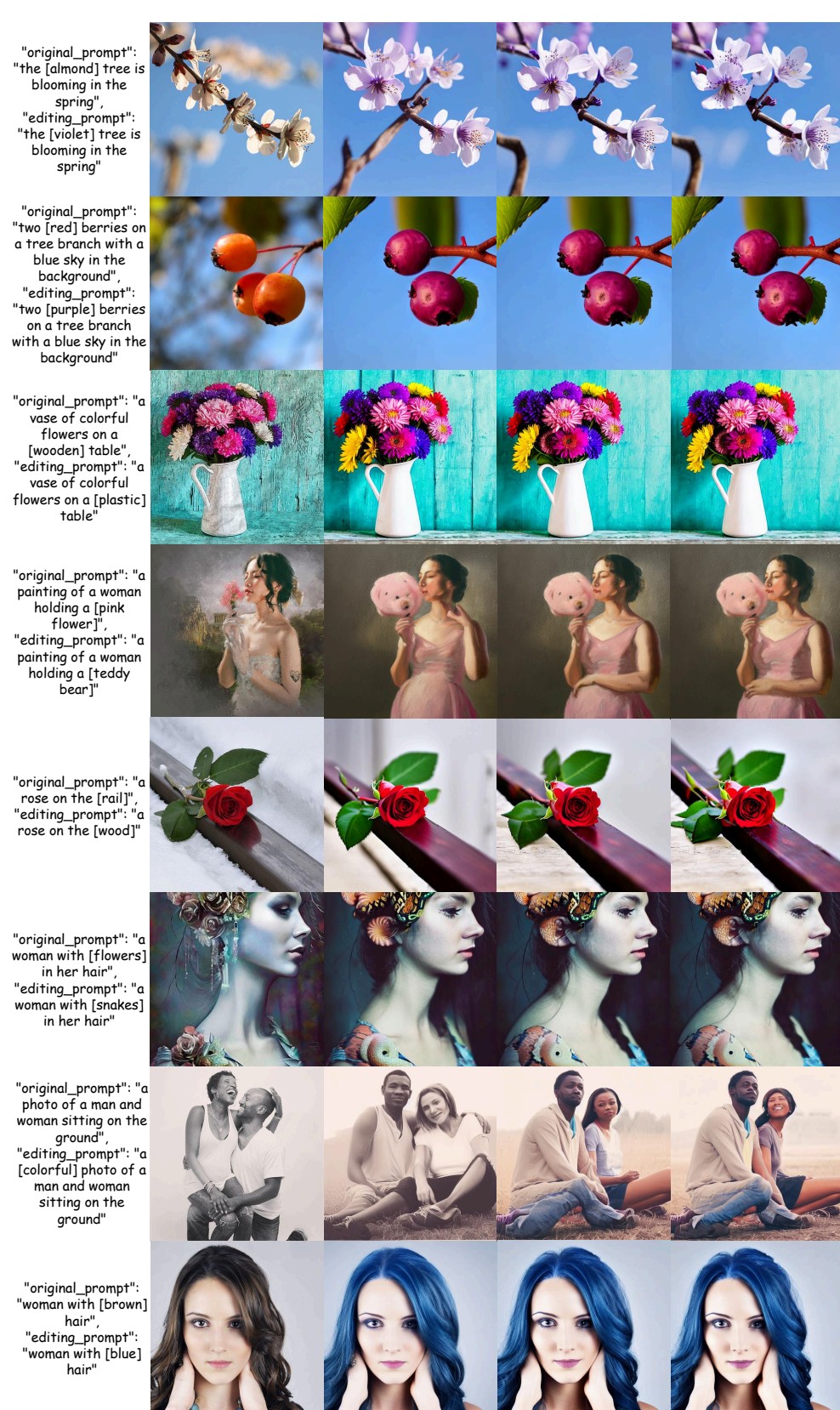

Figure 7: The image editing results in PIE-Bench.

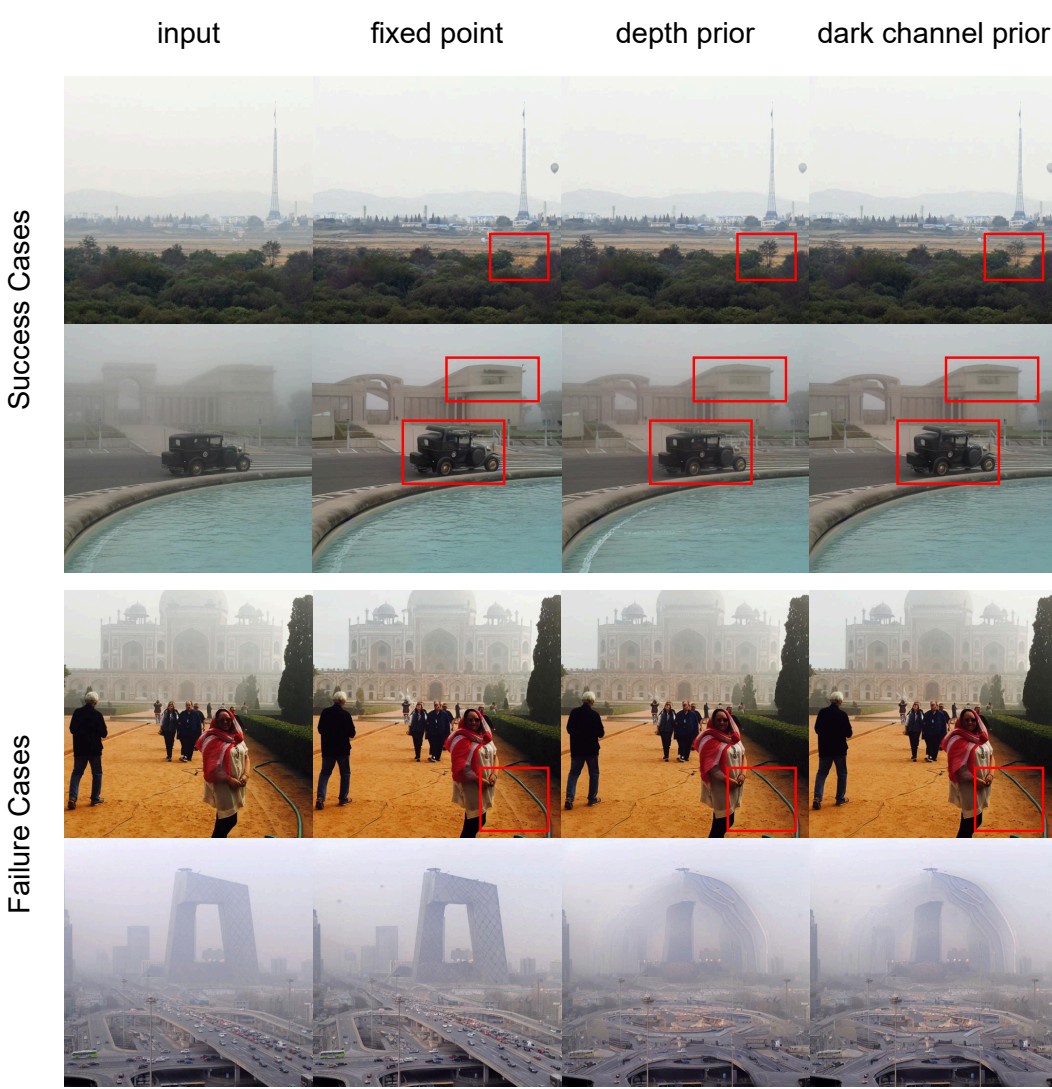

Figure 8: The class-free guidance coefficient is set to 1. The red boxes highlight some noticeable changes, which are beneficial in the success cases but detrimental in the failure cases.

## A.3 Dark Channel and Depth Prior

We present success cases and failure cases of incorporating depth prior and dark channel prior, as shown in Figure 8. In the experiments, we directly utilized the fixed point without NTI. To prevent the collapse caused by an increased class-free guidance coefficient, we set the class-free guidance coefficient to 1 in this case.

