# OpenReview forum: "Exploring Fixed Point in Image Editing: Theoretical Support and Convergence Optimization"
_NeurIPS.cc/2024/Conference — NeurIPS 2024 poster_

### Official Review · Reviewer_eqay · 2024-07-10

**Soundness:** 3
**Presentation:** 3
**Contribution:** 3
**Rating:** 6
**Confidence:** 4

**Summary:**

This paper provides important theoretical support and practical optimization in the field of image editing, especially the fixed point problem in the DDIM inversion process. Through the application in image editing and dehazing tasks, the effectiveness and generalization potential of the fixed point theory are demonstrated. The paper has performed well in theoretical contribution and experimental verification.

**Strengths:**

1. Based on theoretical findings, the authors optimized the loss function in the original DDIM inversion and improved the visual quality of image editing.
2. The fixed point theory is applied to the image dehazing task, and unsupervised dehazing methods based on text guidance are explored, showing the generalization potential of the fixed point theory.
3. Experiments on the PIE-Bench dataset verify the effectiveness of the optimized fixed-point convergence criterion and demonstrate the improved image editing quality.

**Weaknesses:**

1. Although the paper discusses fixed-point computational optimization, no detailed computational resource requirements (i.e. GFLOPs, param.) are provided, which may affect the reproducibility of the experiments.
2. The paper does not report the error margins or statistical significance tests for the experimental results, which limits the interpretability of the results.
3. The experiments mainly focus on image editing and dehazing tasks, and it may be necessary to verify the generalization ability of the fixed point theory in a wider range of image restoration tasks.
4. I am very curious whether the method proposed by the author is as robust to some complex nonlinear degradation as to simple linear degradation.

**Questions:**

I don't particularly understand the specific theoretical derivation in the article, but I think the experimental part still has room for improvement. I will decide my final rating based on the author's rebuttal and the opinions of other reviewers. Overall, I think this article is interesting.

---

> ### Author Rebuttal · Authors · 2024-08-05
>
> We thank Reviewer eqay for devoting time to this review and providing valuable comments.
>
> **Weaknesses:**
> > **W1:** No detailed computational resource requirements (i.e. GFLOPs, param.) are provided.
>
> **A:** Our approach relies on the Stable Diffusion v1.4 model, which is available through the Diffusers library. As such, the GFLOPs and parameter counts can refer to the official documentation of Stable Diffusion v1.4. Of course, we have also calculated the GFLOPs and parameters of each module in Stable Diffusion v1.4 ourselves, and the following data is for reference only:
> | | vae| unet| text_encoder|
> | :---: | :---: | :---: |:---:|
> |parameter| 83.65M| 859.52M | 123.06M  |
> | GFLOPs| 1773.5  | 222.3    |0.12    |
>
> > **W2:** The paper does not report the error margins or statistical significance tests for the experimental results.
>
> **A:** The Reviewer eqay are correct that we should have included these important measures to provide a more rigorous evaluation. Upon your recommendation, we have conducted additional runs and calculated the mean values along with the error margins. The updated results are as follows:
> | | Distance   | PSNR   | LPIPS   | MSE   | SSIM   | Whole   | Edited   |
> | :---: | :---: | :---: |:---:| :---: | :---: | :---: | :---: |
> |Image Editing | 69.181$\pm$0.001 | 17.88$\pm$0.01  | 208.87$\pm$0.02  | 219.86$\pm$0.05    |  71.24$\pm$0.18   | 25.33$\pm$0.016   |  22.59$\pm$0.001  |
> | Image Reconstruction |  70.247$\pm$0.20  | 17.80$\pm$0.07    | 210.70$\pm$0.03     | 225.47$\pm$0.01    |  70.90$\pm$0.05 | 23.81$\pm$0.004    |  21.39$\pm$0.02  |
>
> > **W3:** It may be necessary to verify the generalization ability of the fixed point theory in a wider range of image restoration tasks.
>
> **A:** We have addressed this issue in lines 226-231 of the paper. The primary reason for the relatively poor performance on tasks like rain and snow removal is that the pre-trained diffusion model we utilized is not accurate in capturing these specific types of degradation. The pre-trained Stable Diffusion v1.4 is not sufficiently adept at generating accurate attention maps for these weather-related artifacts. To address this limitation, the solution would be to train a diffusion model that is specifically designed to capture the attention maps corresponding to rain and snow. And it requires the use of shallow, pixel-level attention maps that can better identify the characteristics of these weather-related degradations.
>
> > **W4:** I am very curious whether the method proposed by the author is as robust to some complex nonlinear degradation as to simple linear degradation.
>
> **A:** The key to the robustness against different types of degradations lies in the accuracy of the attention map matching. Based on our experimental observations, Stable Diffusion v1.4 was able to capture the attention maps for degradations such as rain, snow, and fog. However, the issue lies in the fact that the default Stable Diffusion v1.4 model uses deep-level attention maps, which are not precise enough to accurately capture the attention maps for rain and snow. As a result, our method's performance on tasks like snow and rain removal was not as effective as we would have liked. Regarding other types of degradations, such as blur, shadows, occlusions, and noise, we have not yet explored them by using Stable Diffusion v1.4. Therefore, we are uncertain to capture the attention maps for these degradations. To achieve more robust results, we believe the solution would be to retrain a diffusion model that is specifically designed to capture the attention maps corresponding to the various types of degradations.

---

> > ### Comment · Reviewer_eqay · 2024-08-12
> >
> > Thank you to the authors for the rebuttal, which has addressed my concerns to some extent.
> >
> > According to the authors' description, the proposed method seems to focus more on semantic-level processing, while its ability to capture details appears to be insufficient. However, many tasks seem to have a strong semantic demand, such as low-light image enhancement. Have the authors attempted any experiments in this area? Merely providing some analysis without concrete examples seems like a limitation to me. On the other hand, it seems that dehazing tasks are more complex in terms of physical imaging compared to deraining tasks. Why, then, is rain more challenging to address? Have the authors considered the non-uniformity of real-world hazy images?

---

> > > ### Author Response · Authors · 2024-08-12
> > >
> > > We appreciate the reviewer eqay's recognition of our work. Regarding the reviewer's questions, we respond as follows:
> > > - Its ability to capture details appears to be insufficient. Have the authors attempted any experiments in low-light image enhancement area?
> > > Since the pre-trained Stable Diffusion V1.4's the semantic attention map is at deeper layers, its ability to capture fine details is somewhat limited.  We have mainly experimented with removing rain, snow, and haze, and have not tried low-light image enhancement. However, if low-light image enhancement also relies on semantic awareness and the attention map also can be captured, it should be effective.
> > > - Why, then, is rain more challenging to address?
> > > The main reason is that the attention map is based on deeper layers, which results in the attention map having a much smaller size compared to the original image. This makes it difficult to accurately capture raindrops or small snowflakes.
> > > - Have the authors considered the non-uniformity of real-world hazy images?
> > > We believe the reviewer is asking about the scenario where the $t(x) = e^{-kd(x)}$ in the equation $I(x) = J(x)t(x) + A(1-t(x))$ has $k$ as a variable. We have tested this scenario and found it to be effective for removing haze. This haze is generated by equipment in an indoor factory setting, and the purpose is to remove the obstruction caused by these haze conditions. However, since this real-world scenario involves the commercial confidentiality of our collaborators, we did not include it in the paper.

---

### Official Review · Reviewer_Cq8S · 2024-07-12

**Soundness:** 2
**Presentation:** 3
**Contribution:** 2
**Rating:** 5
**Confidence:** 5

**Summary:**

The paper concerns with the fixed points of the DDIM inversion scheme, which is central to many image editing methods. The authors first establish that the DDIM inversion process at any given time step $t$ exhibits a unique fixed point by demonstrating that the corresponding functional has a Lipschitz constant strictly smaller than 1.

Furthermore, the authors reveal that the commonly employed stopping criterion $f(z_t)-z_t$ in the fixed-point inversion process might not guarantee convergence to the optimal solution. To address this, they introduce a new optimization criterion, $f(z_t)-f(\hat z_t)$, where $\hat z_t$ represents a perturbed version of $z_t$,  creating a new optimization path. Since the fixed point is unique, both the original and perturbed paths should converge to the same point. Thus, the new criterion ensures convergence to solutions that better approximate the desired fixed point, yet at the expense of a few additional iterations per time step.
The authors of demonstrate the proposed method through experiments on image editing and unsupervised dehazing tasks.

**Strengths:**

* The paper is well-written. The authors provide a clear motivation and necessary background, and their derivations are easy to follow.
* The contributions of the paper are solid. The presented fixed-point analysis is simple yet insightful, and the new optimization criterion is both creative and easy to implement.
* The experiments, while not extensive, are sufficient to demonstrate the proposed method.

**Weaknesses:**

* While I believe the  the theoretical conclusions derived  in Section 3 are correct, there are several inaccuracies that must be addressed to ensure the soundness of the derivations. Specifically, $z_t$ and $f(z_t)$ are vectors, hence, the inequality in (7) is not clear. How it is defined? It is pointwise ineuqliaty? Furthermore, it seems that the authors assume the difference between the scores on the right-hand side of (7) is non-negative, which needs to be justified.
Continuing in this regard, the differences on both sides of (11) are vector differences, making the inequality unclear. I believe Inequality (11) should be  $||f(z_t^i)-f(z_t^j)||\leq (1-k\sqrt{\bar{\alpha}}) ||z_t^i-z_t^j||$.

* The motivation and intuition behind equation (12) should be explained prior to its presentation. Currently, the purpose of this equation is unclear upon initial reading.

* The implications of inequality (14) should be clarified. Why is this inequality interesting? How does it affect fixed-point convergence, and what role does $\delta$ play in this process? Are there any conditions on $\delta$ that need to be assumed to ensure proper convergence?

* In contrast to previous sections, I find Section 5 to be lacking necessary mathematical derivations, explanations, and background on NTI. In its current form, the section provides only very high-level details, obscuring the paper's contribution in this context.

* In the image editing experiments, the proposed method appears to offer only marginal improvement, both visually and quantitatively. In the dehazing experiments, the method seems to provide a marginal visual improvement, but it does demonstrate a notably more stable recovery process.

**Questions:**

* Please address the weakness above.
* Can the theoretical analysis presented lead to more efficient algorithms than the straightforward Picard iteration method?
* Can any theoretical statements be made regarding the change in convergence rate under the proposed optimization loss?

**Limitations:**

* While I appreciate the extensive discussion on the application of dehazing, I would appreciate further discussion on the proposed underlying method (or optimization loss), which, in my view, is the central focus of the paper.

---

> ### Author Rebuttal · Authors · 2024-08-05
>
> We thank Reviewer Cq8S for devoting time to this review and providing valuable comments.
>
> **Weaknesses:**
> > **W1:** Is Inequality (7) pointwise ineuqliaty? Inequality (7) and (11) are missing $||\cdot||$.
>
> **A:** Inequality (7) is a pointwise inequality. Additionally, Inequalities (7) and (11) should have the $|\cdot|$ notation, similar to Inequality (12). We apologize for the oversight that caused any inconvenience. We have also adopted your suggestion to replace the $|\cdot|$ notation in Inequality (12) with $||\cdot||$, and have added the $||\cdot||$ notation to Inequalities (7) and (11) as well. We are grateful for your careful review and helpful feedback.
>
> > **W2:** The motivation and intuition behind equation (12) should be explained prior to its presentation.
>
> **A:** The intuition behind Inequality (12) is that in the absence of numerical errors, the path $\hat{z}_t$ and the path $z_t$ will converge to the same point, and the difference between the paths $\hat{z}_t$ and $z_t$ will also converge to $0$. The convergence rate of the three is the same in the ideal case.
>
> > **W3:** The implications of inequality (14) should be clarified. What role does $\delta$ play in this process?
>
> **A:** Inequality (14) demonstrates that in the non-ideal case, the convergence of the difference between paths will be slower than the convergence of the difference within a single path. In other words, when using the intra-path loss, while it may appear to have converged, we cannot be certain of the distance from the ideal fixed point.  However, The inter-path loss provides a reference for this. The role of $\delta$ is to reflect the fact that in the presence of numerical errors, the inter-path loss will converge more slowly than the intra-path loss. As a result, the inter-path loss can provide a perspective that is closer to the ideal fixed point. The experiments shown in Figure 1 also validate this point. In general, the $\delta$ value is a very small number. Even in the presence of numerical errors, the function would still converge to a small interval. If the $\delta$ value were to disrupt the contraction mapping property, it would lead to the function failing to converge to a small interval, which would be contradictory to the conclusions proved in Section 3.
>
> > **W4:** I find Section 5 to be lacking necessary mathematical derivations, explanations, and background on NTI.
>
> **A:** We apologize for the lack of sufficient explanation and background regarding the NTI. NTI involves optimizing the null-text embedding as a learnable parameter. Its fine optimization compensates for the inevitable reconstruction error caused by the classifier-free guidance component, and this is crucial for image restoration. From the perspective of the mathematical formulation, the classifier-free guidance prediction is defined as:
> $$
> \tilde{\varepsilon_{\theta}}(z_t,t,C, \varnothing)=w \cdot {\varepsilon_{\theta}}(z_{t}, t, C)+(1-w) \cdot \varepsilon_{\theta}(z_{t}, t, \varnothing)
> $$
> Where $C$ represents the text embedding, $\varnothing$ is the null-text embedding, $z_{t}$ denotes the input at timestep $t$, and $w$ is the guidance scale parameter. The primary optimization target of NTI is the $\varnothing$ term. We hope this provides the necessary background and explanation to clarify the NTI.
>
> > **W5:** In the image editing, the proposed method appears to offer only marginal improvement.
>
> **A:** The marginal improvement in image editing is primarily due to the fact that we chose to use a fixed number of iterations. This may have resulted in some cases where the solution did not fully converge to the ideal fixed point. Nevertheless, there is still room for performance improvement. As mentioned in lines 155-159 of the paper, one potential strategy is to compute multiple inter-path losses and then select the cluster center of the converged points as the final ideal fixed point.
>
> **Questions:**
> > **Q1:** Can the theoretical analysis presented lead to more efficient algorithms than the straightforward Picard iteration method?
>
> **A:** The fixed point computation in this work is based on the Picard iteration method. If more efficient algorithms are desired, we would recommend exploring Aitken's acceleration method or Steffensen's method. Aitken's acceleration can provide faster convergence than the basic Picard iteration by extrapolating the sequence of iterates. Steffensen's method is another technique that achieves higher order convergence compared to the Picard iteration.
>
> > **Q2:** Can any theoretical statements be made regarding the change in convergence rate under the proposed optimization loss?
>
> **A:** Regarding the change in convergence rate, we only provided a rough illustration of the slight increase in the Lipschitz constant using the inequality (14) to demonstrate the change in convergence rate. Given that the observed change in convergence rate in the experiments is subject to the influence of numerical errors, which may vary across different scenarios, we did not provide further theoretical statements. However, we can offer an intuitive statement. If we use the original fixed-point loss, which takes a single-path perspective, the convergence rate would be consistent with the ideal case, as there are no other references. When the intra-path loss changes minimally, it would consider the fixed point to be reached. In contrast, the multi-path perspective introduces the inter-path loss. Even when the intra-path loss changes very little, the inter-path loss may still exhibit a decreasing trend. Furthermore, even if the intra-path loss is zero, due to numerical errors, the inter-path loss may not be exactly zero. This implies that the inter-path loss provides additional information indicating that we still need to travel a small distance to reach the ideal fixed point. As a result, the inter-path loss may converge slightly slower than the intra-path loss in the presence of numerical errors.

---

> > ### Comment · Reviewer_Cq8S · 2024-08-11
> > **Post-Rebuttal**
> >
> > I thank the authors for their rebuttal, which addressed most of my concerns satisfactorily.  However, an issue that remains, and was also raised by other reviewers, is the marginal improvement in image editing quality. The authors attributed this to the use of a fixed number of iterations.
> >
> > My follow-up question is how this specific number of iterations was determined. Was there an ablation study conducted to investigate its impact? Additionally, why not allow the process to run until approximate convergence is achieved, based on some predefined stopping criteria?
> >
> > While I appreciate that the focus of the paper is primarily theoretical, I believe it is crucial for theoretical contributions to demonstrate some practical merits as well. This could include faster convergence, improved stability, or enhanced image quality.

---

> > > ### Author Response · Authors · 2024-08-11
> > >
> > > We appreciate the reviewer Cq8S's recognition of our work. Regarding the reviewer's questions, we respond as follows:
> > > - How was this specific number of iterations determined?
> > > We referred to the strategies used in AIDI and FPI, which observed the loss convergence behavior through some data and then determined a fixed number of iterations. Their observations showed that the intra-path loss can converge in about 3 iterations, so the experiments using intra-path loss used 3 iterations. The inter-path loss was observed to converge in about 6 iterations, so we used 6 iterations. This is a relatively fair configuration.
> > > - Was there an ablation study conducted to investigate its impact?
> > > We experimented with gradually increasing the number of iterations from 3 to 6, as well as decreasing from 6 to 3, in 50 time steps for image editing. The performance was between the 3-iteration and 6-iteration results, with the former performing slightly better than the latter. We had only saved the results for the former in our previous experiments.
> > > |  Distance   | PSNR  | LPIPS  | MSE  | SSIM  | Whole  | Edited  |
> > > |  ----  | ----  | ----  | ----  | ----  | ----  | ----  |
> > > | 69.46  | 17.84 | 208.91 | 220.44 | 71.14 | 25.25 | 22.55 |
> > > - Why not allow the process to run until approximate convergence is achieved, based on some predefined stopping criteria?
> > > We did try setting a threshold to stop the process. However, the threshold that works for different images and time steps can vary, and the manually set thresholds took a long time to reach in some cases. For those that could not reach the threshold, we stopped at 50 iterations. But we found the experimental results on some metrics were not as good as the fixed iteration approach, and it took longer. The main reason was that the manually set thresholds might be higher than the ideal values in some cases. Therefore, we did not mention this strategy in the paper, and instead recommend using the clustering center of multiple path iterations. Considering a fairer comparison with AIDI and FPI strategies, time efficiency, and verifying the defects of intra-path loss, we decided to use the fixed iteration approach.
> > > - It is crucial for theoretical contributions to demonstrate some practical merits as well.
> > > Since the gains of fixed points are limited in image editing, we added extra unsupervised dehazing experiments in the paper to further demonstrate the practical value and potential of fixed points in other tasks.

---

### Official Review · Reviewer_5f4S · 2024-07-12

**Soundness:** 4
**Presentation:** 3
**Contribution:** 3
**Rating:** 5
**Confidence:** 4

**Summary:**

The paper proves the existence and uniqueness of fixed points in DDIM inversion using the Banach fixed-point theorem. It identifies flaws in existing fixed-point loss functions and proposes optimizations to improve convergence and visual quality of edited images. It also introduces a novel text-based approach for unsupervised image dehazing using fixed-point based editing.

**Strengths:**

The paper addresses the inconsistencies in image editing results caused by errors introduced during the DDIM inversion process. It provides a solution to enhance the reliability and quality of edited images with the proposed fixed-point optimization. The paper also gives a thorough analysis of DDIM inversion, identifying key issues and proposing optimization strategies for it. This detailed analysis helps to undetstand the challenges in the context of image editing. The paper also outlines clear directions for future work.

**Weaknesses:**

- the performance gains the paper proposes are minor. It might not be worth the complexity of the approach.
- the paper doesn't seem to have thoroughly tested its method on various challenging situations, especially for image dehazing.
- the paper acknowledges that proposed method may introduce additional computational overhead when optimizing for fixed point convergence. However, specific quantitative measures of this overhead are not provided, so further analysis is needed.

**Questions:**

how does the proposed fixed-point based dehazing approach compare with state-of-the-art dehazing techniques in terms of performance and computational efficiency?

**Limitations:**

The authors have addressed the limitations of their work, along with potential negative societal impacts.

---

> ### Author Rebuttal · Authors · 2024-08-05
>
> We thank Reviewer 5f4S for devoting time to this review and providing valuable comments.
>
> **Weaknesses:**
> > **W1:** The performance gains the paper proposes are minor.
>
> **A:** The reason why the performance gains are minor in image editing is that across the entire dataset, we used a fixed number of iterations to reach the fixed-point. This may have resulted in some images not converging fully at certain time steps. It's important to note that the focus of this paper is on providing theoretical support for the fixed-point and optimizing the fixed-point loss. Therefore, using a fixed set of variables for the experiments is more equitable. However, in the case of unsupervised image dehazing, the performance gains are significant, as shown in the last two columns of Figure 5 in the paper. The use of the fixed-point effectively avoids the collapse of the recovered images.
>
> > **W2:** The paper doesn't seem to have thoroughly tested its method on various challenging situations.
>
> **A:** In terms of fixed-point based image restoration, we have only conducted experiments on image dehazing, and the specific reasons for this are explained in lines 226-231 of the paper. The reason is that haze, as a pervasive degradation, is relatively easier to capture the corresponding attention map at the semantic level, as shown in the last row of Figure 5 in the paper. Capturing the semantic-level attention map is heavily dependent on the pre-trained diffusion model used, and in this work, we have relied on the Stable Diffusion v1.4 model from the diffusers library. We have also experimented with Stable Diffusion v1.4 for image desnowing and deraing, but found that it was unable to accurately capture the attention maps corresponding to the rain and snow regions. Therefore, we believe that to extend the fixed-point based image restoration to other degradations, a diffusion model specifically trained to capture the attention maps for the corresponding degradations would be required. Finally, the primary focus of this work is to provide the theoretical support for the existence and uniqueness of the fixed-point, as well as the optimization of the fixed-point loss. We have also aimed to demonstrate the potential of the fixed-point in image restoration tasks, as exemplified by the image dehazing results.
>
> > **W3:** Specific quantitative measures of this overhead are not provided, so further analysis is needed.
>
> **A:** The Reviewer 5f4S make a valid point regarding the additional computational overhead. We have therefore included the runtime information for reference. The additional computational overhead is primarily incurred during the DDIM inversion process. Specifically:
>   (1) The original fixed-point loss takes approximately 0.26s per individual time step.
>   (2) The improved fixed-point loss takes approximately 0.61s per individual time step.
>   (3) The total time to edit a single image using the original fixed-point loss is approximately 41s.
>   (4) The total time to edit a single image using the improved fixed-point loss is approximately 60s.
>
> In both cases, the total number of time steps is 50. We hope this additional information helps to address your concern regarding the computational overhead.
>
> **Questions:**
> > **Q1:** How does the proposed fixed-point based dehazing approach compare with state-of-the-art dehazing techniques.
>
> **A:** Since the focus of this paper is on the theoretical support and convergence optimization, the performance is lagging behind the state-of-the-art dehazing techniques. The main reason for this performance gap is the inherent limitation of the pre-trained diffusion models in accurately capturing the attention maps corresponding to the degradations. As shown in the last row of Figure 5, the haze attention map is not entirely precise. To address this issue, we would need to utilize lower-level attention maps and train an additional diffusion model that can accurately capture the degradation-specific attention maps. In this work, our aim has been to demonstrate the potential of the fixed-point based image restoration approach in terms of its semantic-level interpretability, which we believe is a key advantage over other methods. Moreover, another key advantage is that with the ability to precisely capture the corresponding degraded attention maps, our method can be applied to arbitrary datasets without the need for additional training.

---

### Official Review · Reviewer_cPeB · 2024-07-13

**Soundness:** 2
**Presentation:** 2
**Contribution:** 1
**Rating:** 3
**Confidence:** 4

**Summary:**

Recent methods treat each step of DDIM inversion as a fixed-point problem to reduce errors, but they lack theoretical support. This paper addresses this gap by making the following contributions: This paper theoretically proves that the Lipschitz constant in DDIM inversion is less than one. By applying the Banach fixed-point theorem, it establishes the existence and uniqueness of fixed points, thus providing the necessary theoretical foundation for image editing methods involving implicit functions. It leverages the uniqueness of fixed points to highlight flaws in existing fixed-point loss methods through theoretical analysis and experimental cases, subsequently proposing optimizations. Furthermore, this paper extends the fixed-point based image editing approach to the task of unsupervised image dehazing, and explores the feasibility of text-guided unsupervised dehazing through fixed-point based editing.

**Strengths:**

1. This paper proves the existence and uniqueness of fixed points in DDIM inversion using the Banach fixed-point theorem.
2. It identifies flaws in the loss functions AIDI and FPI used during inversions.
3. The authors modify the loss function to $f(z_t) - f(\hat{z}_t)$

**Weaknesses:**

1. In Section 3, the statement "Due to the fact that different initial values at time step $t$ can lead to similar $z_0$, i.e., any position at time step $t$ can converge to a small range defined by the rough estimation of $z_0$, we can obtain $|\epsilon_0| = k|\epsilon_t|$, where $0 < k < 1$," might be assertive. This assertion is especially unconvincing when $t$ is large. A rigorous proof for this part would be beneficial.

2. The experiment only compares NTI and P2P, omitting the results of AIDI and FPI. A more extensive and fair comparison should be conducted. Additionally, the image editing results are unsurprising due to the nature of the inversion-based method, and the human editing results appear to be tweaked.

3. The optimization for the image editing task occurs only during inversion. However, this method inherently offers less controllability over the output compared to the input. How does this method address this limitation?

**Questions:**

My question is included in the weaknesses section. I would be willing to raise the rating if my concern is addressed.

**Limitations:**

Please check weakness

---

> ### Author Rebuttal · Authors · 2024-08-05
>
> We thank Reviewer cPeB for devoting time to this review and providing valuable comments.
>
> **Weaknesses:**
> > **W1:** This assertion is especially unconvincing when $t$ is large.
>
> **A:** The premise of this assertion is the DDIM deterministic sampling without perturbation. Under this premise, it is theoretically the case that any position at any time step t  will converge to the same position, which implies that the mean of $\varepsilon_0$ is 0 and the variance is very small. This property also allows the diffusion model to be generalized to other tasks such as image restoration[1],[2]. Therefore, the first half of the assertion is valid.  Regarding the statement "$|\varepsilon_0| = k|\varepsilon_t|$, where $0< k < 1$", we can provide a proof by contradiction. Suppose $k \geq 1$, and when $t$ is large, we let $t = T$. In this case, ${\varepsilon_t} \sim \mathcal{N}(0, 2I), {\varepsilon_0} \sim \mathcal{N}(0, 2k^{2}I)$, which implies that the variance of the distribution of $\varepsilon_0$ is greater than $2I$, which clearly contradicts the premise. Furthermore, ${\varepsilon_0} \sim \mathcal{N}(0, 2k^{2}I)$ means that the pre-trained diffusion model is not a convergent model. Hence, the assertion is valid.
> *References:*
> [1].Wang J, Yue Z, Zhou S, et al. Exploiting diffusion prior for real-world image super-resolution[J]. International Journal of Computer Vision, 2024: 1-21.
> [2].Sun H, Li W, Liu J, et al. Coser: Bridging image and language for cognitive super-resolution[C]//Proceedings of the IEEE/CVF Conference on Computer Vision and Pattern Recognition. 2024: 25868-25878.
>
> > **W2:** The experiment only compares NTI and P2P, omitting the results of AIDI and FPI. The image editing results are unsurprising.
>
> **A:** Our response to this question is as follows:
> 1. The purpose of this paper is to provide theoretical support for AIDI and FPI, and to improve the fixed-point loss they use. The relationship between this work and theirs is complementary, not competitive. To achieve better performance in image editing, AIDI uses Blended Guidance and Stochastic Editing, while FPI uses Prompt-aware Adjustment. However, neither AIDI nor FPI have open-sourced their code, and reproducing these methods is not the focus of this paper.
>
> 2. "the image editing results are unsurprising" is due to the fact that we used the same number of iterations for all images at different time steps, which may have led to some images not converging fully at certain time steps. However, this is still sufficient to demonstrate that the fixed-point loss used by AIDI and FPI is not perfect, and that there is room for improving the performance of AIDI and FPI. This paper also provides suggestions on how to further improve the performance, as mentioned in lines 155-159, where it states that if we can find the cluster centers, the performance will be further improved. Finally, the key point of this work is meant to provide theoretical support for AIDI and FPI and improvements to the fixed-point loss used by AIDI and FPI, rather than to compete with their methods. We hope this clarifies the relationship and the purpose of this work.
>
> > **W3:** This method inherently offers less controllability over the output compared to the input.
>
> **A:** The main application of the fixed-point loss is in DDIM inversion, and this is also the focus of this work. If one wishes to have more control over the output, the Blended Guidance method used in AIDI could be a suitable approach. The Blended Guidance method adjusts the guidance scale during sampling. It increases the guidance scale for regions corresponding to the positive attention map, and decreases the guidance scale for regions corresponding to the negative attention map. This allows for more modifications to the areas that need to be edited, while making fewer changes to the areas that do not require editing.

---

### Decision · Program_Chairs · 2024-09-25

**Decision:**

Accept (poster)

**Comment:**

This paper received a Weak Accept, 2 Borderline Accept and a Reject. The reviewers found the theoretical contributions of the paper interesting. They also raised a number of concerns: 1) Some missing details in the analysis and 2) limited experiments and comparisons.
The authors engaged in discussions with the reviewers and most reviewers found the answers satisfactory. One reviewer is still concerned with a potential limitation in the theoretical analysis. The AC has read all the reviews and the discussion. The AC finds that the authors have addressed most concerns and recommends the paper for acceptance.